# Cold Atmospheric Plasma Jet Treatment Improves Human Keratinocyte Migration and Wound Closure Capacity without Causing Cellular Oxidative Stress

**DOI:** 10.3390/ijms231810650

**Published:** 2022-09-13

**Authors:** Aurélie Marches, Emily Clement, Géraldine Albérola, Marie-Pierre Rols, Sarah Cousty, Michel Simon, Nofel Merbahi

**Affiliations:** 1Toulouse Institute for Infectious and Inflammatory Diseases (Infinity), Toulouse University, CNRS, Inserm, Université Paul Sabatier, Place du Dr Baylac, 31059 Toulouse, France; 2Laboratoire des Plasmas et Conversion d’Énergie (LAPLACE), 118 Route de Narbonne, 31062 Toulouse, France; 3Institut de Pharmacologie et de Biologie Structurale (IPBS), 205 Route de Narbonne, 31077 Toulouse, France; 4UFR Odontologie, CHU Toulouse, 3 Chemin des Maraîchers, 31400 Toulouse, France; 5Département de Chirurgie et Médecine Orales, Centre Hospitalo—Universitaire de Toulouse, 3 Chemin des Maraîchers, CEDEX 9, 31059 Toulouse, France

**Keywords:** keratinocytes, cold atmospheric plasma, oxidative stress, reactive oxygen and nitrogen species, cell migration, skin, wound healing

## Abstract

Cold Atmospheric Plasma (CAP) is an emerging technology with great potential for biomedical applications such as sterilizing equipment and antitumor strategies. CAP has also been shown to improve skin wound healing in vivo, but the biological mechanisms involved are not well known. Our study assessed a possible effect of a direct helium jet CAP treatment on keratinocytes, in both the immortalized N/TERT-1 human cell line and primary keratinocytes obtained from human skin samples. The cells were covered with 200 µL of phosphate buffered saline and exposed to the helium plasma jet for 10–120 s. In our experimental conditions, micromolar concentrations of hydrogen peroxide, nitrite and nitrate were produced. We showed that long-time CAP treatments (≥60 s) were cytotoxic, reduced keratinocyte migration, upregulated the expression of heat shock protein 27 (HSP27) and induced oxidative cell stress. In contrast, short-term CAP treatments (<60 s) were not cytotoxic, did not affect keratinocyte proliferation and differentiation, and did not induce any changes in mitochondria, but they did accelerate wound closure in vitro by improving keratinocyte migration. In conclusion, these results suggest that helium-based CAP treatments improve wound healing by stimulating keratinocyte migration. The study confirms that CAP could be a novel therapeutic method to treat recalcitrant wounds.

## 1. Introduction

Skin wound healing is a complex and organized series of sequential yet overlapping processes induced to restore tissue integrity and homeostasis (Appendix A). Healing involves a hemostasis/coagulation phase in parallel with an inflammatory phase with recruitment of neutrophils, monocytes and macrophages to the site of injury, a proliferative phase affecting fibroblasts, keratinocytes and endothelial cells, and a remodeling phase during which the wound contracts through reorganization of the extracellular matrix [1,2]. Reactive oxygen and nitrogen species (RONS) are key players in these wound healing processes as they control cell functionality and wound phase transitions [3]. Keratinocytes also play a crucial role since they are responsible for restoring the epidermis through re-epithelialization. To achieve this goal, they undergo important phenotypic changes, in particular an increase in their proliferation and migration, and modification of their differentiation program [4]. Chronic wounds fail to progress through these stages in an orderly manner [2], in particular, the re-epithelialization process is impaired [4]. Chronic wounds often have excessive levels of pro-inflammatory cytokines and proteases, leading to a proteolytic and degrading wound environment, reduced mitogenic activity and persistent infections [5,6].

Physical plasma is a particular state of ionized and excited gas molecules. In the 1990s, technologies were developed to generate plasma at low temperature and under atmospheric pressure, called Cold Atmospheric Plasma (CAP). This development stimulated an active field of research for possible biological effects of plasma [4,7,8], whose main effector is the variety of resulting RONS. Since CAP has been shown to display antimicrobial activity against a broad spectrum of microorganisms [9,10,11], and to stimulate the proliferation of several cell types in vitro [12,13,14,15], and because many wound healing processes are controlled by redox reactions, CAP treatments have been proposed as a therapeutic strategy for chronic wounds [16,17]. Only a few studies have shown that CAP promotes re-epithelialization and acceleration of wound closure in vivo in animal models [7,8,14,18] and in human clinical trials [5,19,20,21]. However, the risks incurred, the types of cells affected, and the biological mechanisms involved are not well known. Contradictory results have been obtained depending on the treatment parameters and the settings of the plasma devices. For example, some studies showed that CAP treatment increases the proliferation and motility of both fibroblasts and keratinocytes [12,14,22], while other studies observed no effects on keratinocytes but only improved fibroblast migration [23]. Shi et al. suggested the effect of CAP is dose-dependent, i.e., low doses improve fibroblasts viability and collagen synthesis while high doses inhibit them [24]. What is more, opposite effects of CAP have been observed on normal versus keloid fibroblasts. In addition, most studies on human keratinocyte response to CAP treatments were performed using the HaCaT and the HPV-immortalized cell lines in 2D cultures [25,26,27,28,29,30,31] while only a few used skin primary keratinocytes [32,33,34] and either 3D or skin explant cultures [33,35].

The purpose of the present study was to decipher the effects of the direct helium plasma jet treatment on keratinocytes using both the immortalized N/TERT-1 human cell line (with conserved differentiation capacities close to those of interfollicular keratinocytes [36]) and primary normal human keratinocytes obtained from abdominal skin samples. We analyzed the possible effects of CAP treatments on keratinocyte viability, proliferation, migration, differentiation, and the capacity to form a 3D reconstructed epidermis. Finally, we evaluated putative CAP-induced cellular stresses, including heat shock and oxidative stress, and mitochondrial dysfunctions.

## 2. Results

### 2.1. Short-Time CAP Treatments Were Non-Cytotoxic and Did Not Influence N/TERT-1 Keratinocyte Proliferation

To study the effects of CAP on keratinocytes, we used our helium jet device based on dielectric barrier discharge in the experimental conditions previously described (applied voltage, frequency, pulse duration and gas flow) [37,38] and with the same distance (2 cm) between the outlet of the plasma jet tube and the surface of the cell cultures (Appendix A). Preliminary experiments showed that, to survive, cells needed to be covered by a small volume of PBS containing 0.9 mM CaCl_2_ and 0.49 mM MgCl_2_ during the treatment. Since the amount of RONS produced by CAP depends on the liquid used, the concentrations of hydrogen peroxide (H_2_O_2_), nitrites (NO_2_^−^) and nitrates (NO_3_^−^) present in the PBS were quantified immediately after exposure for increasing periods of time, ranging from 10 s to 120 s (Appendix A). As expected, we observed a time-dependent statistically significant increase in the three RONS species, when compared to a 120 s exposure to helium gas only without any electrical excitation (discharge off) used as control. The concentration of H_2_O_2_ increased from 1.71 ± 0.45 µM after 10 s exposure to the plasma, to 11.47 ± 1.88 µM after 120 s exposure. The concentration of nitrites increased from 23.78 ± 3.12 µM after 20 s exposure to 40.93 ± 5.74 µM after 120 s exposure. The increase in the concentration of nitrates became statistically significant from 40 s exposure on (22.11 ± 2.56 µM) and reached 68.22 ± 6.20 µM after the longest exposure (120 s).

First, we investigated a possible cytotoxic effect of the CAP treatment using the N/TERT-1 human keratinocyte cell line. The cells were treated directly for increasing periods of time (10–120 s) or left untreated, then returned to fresh untreated culture medium, and viability was quantified 1 h, 24 h and 48 h after plasma treatment using the MTT assay (Figure 1A). The optical density corresponding to control untreated cells increased over the 2-day culture period. Long treatments (60 s and 120 s), inhibited keratinocyte proliferation, whereas CAP treatments of 50 s and less did not affect the keratinocyte proliferation rate (Figure 1A). Using immunocytochemistry targeting the proliferating cell protein marker Ki67 (Appendix A), we confirmed that CAP treatments of N/TERT-1 keratinocytes of less than 60 s did not affect proliferation whereas longer treatments did. Indeed, 120 s treatments reduced the proportion of proliferating cells (0.71 ± 0.05) measured 48 h post-treatment. Since CAP has been reported to induce apoptosis [39,40,41], we detected double-strand DNA breaks generated during apoptosis using the TUNEL assay 24 h after CAP treatment (Figure 1B). Only the longest treatment (120 s) triggered an increase in the number of apoptotic cells, further confirming the non-deleterious effect of light plasma treatment of N/TERT-1 keratinocytes in our experimental conditions.

We then used non-cytotoxic conditions (60 s treatment and below) to test for possible plasma-induced stress, in particular because CAP is known to release heat (approx. 40 °C). Induction of the heat shock protein 27 (HSP27) was analyzed by immunofluorescence. One hour and 24 h after exposure to plasma, the shorter 20 s CAP treatment did not induce a significant number of HSP27 positive cells compared to the control gas condition (Figure 1C). However, a significant increase in the number of HSP27 positive cells was observed with the 40 s and 60 s CAP treatments.

### 2.2. Short CAP Treatments Did Not Affect N/TERT-1 Mitochondria and Intracellular ROS Levels

CAP has been suggested to induce apoptosis in cancer cells via a significant increase in intracellular ROS [16,39] as well as mitochondrial damage including alteration of the mitochondrial transmembrane potential [42]. To assess changes in the mitochondrial function in CAP treated N/TERT-1 keratinocytes, we used the red-fluorescent probe TMRM that localizes in mitochondria and allows to detect mitochondrial membrane depolarization in living cells using confocal analysis (Figure 2A,B). Treatment of cells with CCCP to induce mitochondrial membrane depolarization, used as a positive control, led to a marked decrease (64.2%) in the fluorescence signal. CAP treatments lasting between 10 s and 60 s induced no significant changes in TMRM labelling. Flow cytometry analysis confirmed this result (Appendix A), strongly suggesting that CAP did not affect the mitochondrial membrane potential. To confirm the absence of CAP induced mitochondrial dysfunction, we used the MitoSOX Red fluorescent probe (Figure 2C,D), which selectively targets mitochondria and is specifically sensitive to the mitochondrial superoxide in living cells, but not to other RONS. Antimycin, a drug known to increase the formation of mitochondrial superoxide, was used as positive control, and indeed, confocal microscopy analysis revealed a marked increase (226.8%) in the fluorescence signal. Six hours after the 40 s and 60 s CAP treatments, the MitoSOX signal was significantly higher than in the control gas alone condition. This indicated over-production of mitochondrial superoxide, since the MitoSOX signal was clearly co-located on the mitochondria, as depicted using the MitoTracker green pattern of fluorescence, suggesting putative oxidative stress and cellular damage induction. In contrast, shorter time plasma treatments (30 s and less) did not induce a significant increase in labelling compared to the control gas condition. Flow cytometry analysis confirmed these results (Appendix A). To assess induction of intracellular ROS in CAP-treated N/TERT-1 keratinocytes, we used the green Cell-ROX ROS detection reagent quantified by flow cytometry (Figure 2E). Long treatments (60 s and 120 s) significantly increased Cell-ROX intensity 24 h after CAP treatment (141% and 213%, respectively), indicating production of intracellular ROS. However, short CAP treatments (20 s and 40 s) did not produce any significant changes in Cell-ROX intensity up to 24 h post-treatment.

### 2.3. Short CAP Treatments of N/TERT-1 Keratinocytes Accelerated Wound Closure In Vitro in a Time-Dependent Manner

Keratinocyte migration is a key step in the re-epithelialization process during wound healing. Consequently, we investigated the effects of our helium plasma device on cell motility using the N/TERT-1 cells and an in vitro wound healing assay, also known as scratch assay. An 800-µm wide standardized scratch was made on a completely confluent cell monolayer, cells were CAP treated, and images of the wound area were taken at hourly intervals for 24 h (where the wound area is completely closed in the control condition) to quantitatively analyze cell migration dynamics with the IncuCyte ZOOM System using bright-field microscopy (Figure 3). N/TERT-1 keratinocytes treated with CAP for 10 s and 20 s presented a scratch wound closure kinetics similar to that in the control gas alone condition. However, 30 s and 40 s exposure times significantly enhanced closure speed and therefore probably cell motility since, as shown above, cell proliferation was not affected. A CAP treatment of 50 s did not produce any effects, whereas a 60 s treatment significantly delayed closure of the scratch wound.

### 2.4. CAP Treatments Had Similar Effects on N/TERT-1 and Human Primary Keratinocytes: Short Treatments Were Not Cytotoxic but Stimulated Cell Migration

In the second part of our experiments, we completed our investigation of the effects of exposure to a helium plasma jet on human keratinocytes using primary keratinocytes isolated from the abdominal skin of three healthy females. The viability of the keratinocytes was evaluated using the MTT assay, 1 h, 24 h and 48 h after CAP treatments lasting between 10 s and 180 s (Figure 1D). Keratinocyte viability 1 h post-treatment was not affected by exposure to CAP whatever the length of the treatment. On a longer term (24 h and 48 h), exposures lasting 20 s and 40 s were not cytotoxic and did not alter keratinocyte proliferation, whereas longer exposures, i.e., 60 s and more, were cytotoxic and reduced cell viability and proliferation. This confirmed the absence of a deleterious effect of short-time plasma treatments on human keratinocytes.

Next, we checked the potential of our plasma device to stimulate the motility of primary keratinocytes. We performed a wound healing assay, in which an ~800 µm wide scratch was manually made with a 200 μL plastic tip on a cell monolayer at complete confluence. After plasma treatment for 10 s to 60 s, the plate wells were monitored by bright-field microscopy at 24 and 48 h (Figure 4A). We observed that short CAP treatments (10 s and 20 s) significantly enhanced keratinocyte migration, while longer treatments had no effect on scratch wound closure. To confirm that the acceleration of wound closure was due to an increase in cell migration but not to an increase in cell proliferation, we repeated the same assay while inhibiting cell proliferation with mitomycin C incubation prior to exposure to CAP (Figure 4B). We obtained strikingly similar results, with 10 s and 20 s CAP treatments significantly increasing wound closure compared to the control. We thus confirmed that CAP treatments accelerate wound closure in vitro by the stimulating cell migration.

### 2.5. CAP Treatment Did Not Affect Primary Keratinocyte Differentiation

The ability of keratinocytes to differentiate and form a neo-epidermis is essential to the re-epithelialization process. We therefore studied the conservation of keratinocyte differentiation potential in vitro after plasma treatments, using a 3D reconstructed human epidermis (RHE) model. The primary keratinocytes were exposed to CAP for 20 s, 40 s and 60 s prior to being cultivated for 12 days at the air-liquid interface, allowing the keratinocytes to stratify, differentiate and form an epithelium close to the interfollicular epidermis. When the resulting epidermis were stained with hematoxylin and eosin (H&E) (Figure 5A), we observed no alterations in their morphology regardless of the length of the CAP treatments. We further analyzed the expression of differentiation markers such as keratin 10, involucrin, filaggrin and corneodesmosin by indirect immunofluorescence (Figure 5B). As expected, in the RHEs produced with gas-only treated keratinocytes, keratin 10 was detected in all suprabasal cells, whereas the three late-differentiation markers were only detected in the most differentiated cells. No changes were observed in any of the labeling patterns after CAP treatments (Figure 5B). In addition, quantification showed that labelling intensity was identical (Figure 5C). These outcomes show that short plasma treatments did not affect the primary keratinocyte differentiation potential.

## 3. Discussion

Clinical trials in human and experimental studies in rodents have shown a beneficial effect of CAP treatments on skin wound healing, but the biological reasons were not fully understood. In this study, we focused on keratinocytes as they are the first skin cells affected by CAP application, but the effects of CAP on these cells are currently relatively understudied. 2D cultures of the HaCaT cell line is by far the most commonly used in vitro model to study keratinocyte response to CAP treatment [18,25,26,27,29,43,44], although HaCaT cells differ from normal keratinocytes in many aspects. Very few studies have been performed using primary human keratinocytes [32,33,34]. We consequently decided to evaluate the effects of direct helium plasma jet treatments on human keratinocytes using both the immortalized N/TERT-1 cell line and primary keratinocytes. Indeed, when cultured at the air-liquid interface, N/TERT-1 cells can form a fully stratified epithelium that is very similar to a real epidermis, with normal stratum corneum permeability, epidermal morphology, responses to inflammatory cytokines and expression of epidermal differentiation markers. In addition, when used in 2D cultures, N/TERT-1 cells also behave like primary human keratinocytes [36]. As far as we know, no studies have been conducted using the N/TERT-1 cell line until now.

In our experimental conditions, we obtained similar results with the N/TERT-1 and primary keratinocytes. First, we demonstrated that short CAP treatments (less than 1 min) were not cytotoxic, did not induce heat-stress and did not affect keratinocyte proliferation and differentiation. Therefore, we assume that calcium chloride and magnesium chloride concentrations were not changed. Using the argon-based kINPenMED medical device to generate CAP, S Hasse et al., showed that CAP treatments of human skin biopsies lasting 1–5 min did not modify keratin 1 and 14 expression, while only the longest exposure did induce apoptosis of the basal keratinocytes [12], in line with our results. Treatment of human primary keratinocytes for 2 min using the MicroPlaSter argon-based device did not cause apoptosis, and did not alter their proliferation rate [45]. The same was shown for in vivo treated skin of mice [45]. In contrast, other results showed a CAP induced increase in keratinocyte proliferation, that varied with the duration of treatment and input power [12,31]. So far it is not known what causes the proliferation of keratinocytes after CAP exposure. The impact of RONS has been suggested, although it should be noted that comparing plasma devices and experimental setups is complex due to the remarkable variability of possible configurations, for example, differences in the type of discharge, the operating gas, flow rate or voltage intensity [37,46,47,48]. With our CAP device and our experimental setup, smaller amounts of RONS were produced than in other studies in which either argon or helium plasma was used, which may explain the differences between the results of our study and those of other studies.

In our experimental conditions, CAP increased keratinocyte migration. The same effect was observed when cells were pretreated with mitomycin C to inhibit their proliferation. Enhanced migration of keratinocytes may well explain the accelerated wound closure observed in vivo. These results are consistent with those of other studies on HaCaT cells [12,14,26,47] as well as HPV-immortalized keratinocytes [31]. Although the mechanism through which plasma influences keratinocyte migration is not well understood, effects on junction and adhesion proteins, with down regulation of E-cadherin and several integrins along with reorganization of the actin cytoskeleton, have been suggested [34,49,50].

Our results suggest that CAP treatments lasting less than 60 s do not induce changes in mitochondria nor in intracellular ROS levels. The absence of observed effects could be due to the short exposure time and the limited amounts of ROS produced in our treatment conditions. It has already been reported that short treatments (<60 s) do not affect intracellular ROS, while longer exposure times (e.g., 300 s [49] and 10–20 min [43]) were reported to cause a significant rise in their levels in HaCaT keratinocytes [27,43,49]. Long exposure times were cytotoxic and induced the breakdown of the mitochondrial membrane potential (an early sign of apoptosis).

Diverse cell types are involved in wound healing, such as fibroblasts synthetizing new extracellular matrix components to restore connective tissue. It has been reported that 60 s of treatment with the He-DBD device increased the proliferation of human fibroblasts [15]. Treatments of human fibroblasts for 30 s using the MicroPlaSter device transiently increased the migration of fibroblasts [23]. Similar results were obtained using the kINPen for 60 s [14]. CAP has also been shown to activate fibroblasts, and to stimulate the production of wound healing-relevant cytokines and growth factors [13,45]. Endothelial cells are responsible for angiogenesis, a crucial physiological wound healing process [51]. CAP has been shown to activate angiogenesis: the formation of pseudo-vessel tubes was improved in vitro when the endothelial HSkMEC.2 cell line was treated with He-DBD for 30 s [15]. Similar results were obtained using the MicroPlaSter device, and 30 s treatments of HUVEC demonstrated pro-angiogenic effects in both autocrine and paracrine modes [32].

In conclusion, our results suggest that helium-plasma jet improves wound healing partly by stimulating keratinocyte migration, an increase in their proliferation is not indispensable. Further work is now need to decipher the mechanisms involved at the cellular and molecular scale, in addition to the suspected role of NRF2/KEAP1 pathway activation [17,52] as well as electrical field induced by CAP production.

## 4. Materials and Methods

Plasma jet device. The helium plasma jet device consisted of a dielectric barrier discharge structure, that is described in detail elsewhere [37,38]. Briefly, two aluminum electrodes were wrapped around a quartz tube and connected to a high-voltage mono-polar square pulses generator. The characteristics of the power supply were 10 kV voltage, 10 kHz frequency and 1 µs pulse duration. Helium gas was delivered through the quartz tube at a controlled flow rate of 3 L.min^−1^. All these operating parameters were kept unchanged in our study.

Cell culture of human N/TERT-1 keratinocytes and human primary keratinocytes. Human N/TERT-1 keratinocytes were kindly provided by JG Rheinwald [53] and cultivated in DermaLife K Complete Medium (Lifeline Cell Technology, San Diego, CA, USA) with 100 U/mL of penicillin, and 100 µg/mL of streptomycin (Gibco, ThermoFisher Scientific, Waltham, MA, USA) at 37 °C and 5% CO_2_ in a humidified atmosphere. The medium was renewed every two days. Primary normal human keratinocytes were isolated from samples of abdominal skin from 3 females (35–54 years old) provided by Genoskin (Toulouse, France) following written informed consent of the donors, and the approval of the French Ministry of Research (#AC-2017-2897). The primary keratinocytes were grown in DermaLife K Complete Medium (Lifeline Cell Technology) with 100 U/mL of penicillin, and 100 µg/mL of streptomycin at 37 °C and 5% CO_2_ in a humidified atmosphere. The medium was renewed every two days.

Keratinocyte exposure to cold atmospheric plasma. Twenty-four hours prior to the experiment, 1 × 10^5^ cells were seeded in a 48-well plate and cultured for 24 h. Direct plasma treatment consisted of exposing the cells covered by 200 µL of Dulbecco’s Phosphate Buffered Saline (DPBS) containing 0.9 mM calcium chloride and 0.49 mM magnesium chloride (Sigma-Aldrich, Saint-Louis, MO, USA) (corresponding to a 2.1 mm thick PBS layer) to the plasma jet effluent at a distance of 2 cm from the surface of the liquid. Exposure times ranged from 10 s to 120 s. To exclude effects of the gas itself, 120 s exposure to the helium gas alone without any electrical excitation (discharge off) but the same other conditions was used as control. Immediately after exposure to CAP, DPBS was replaced with fresh culture medium, and the keratinocytes were further cultured.

Cell viability assay. Cell viability was assessed using the 3-(4,5-dimethylthiazol-2-yl)-2,5-diphenyltetrazolium bromide (MTT) assay (Sigma-Aldrich). Briefly, MTT was added at a final concentration of 0.5 mg/mL to the culture plates 1 h, 24 h and 48 h after the CAP treatments. After 3 h of incubation, the supernatants were aspirated. The formazan crystals in each well were dissolved in dimethyl sulfoxide, and absorbance at 570 nm was measured using the Biochrom Asys UVM340 plate reader (Biochrom Ltd., Cambridge, UK).

DNA fragmentation assay. Apoptotic DNA fragmentation was evaluated with a terminal deoxynucleotidyl transferase dUTP nick end labeling (TUNEL) assay (Invitrogen, ThermoFisher Scientific, Waltham, MA, USA) according to the manufacturer’s protocol. Cells grown on glass coverslips for 24 h were exposed to plasma. One µM staurosporine (Sigma-Aldrich) was used as a positive control for DNA double strand breaks. Twenty-four hours later, cells were washed with PBS, fixed with 1% (*w*/*v*) paraformaldehyde, permeabilized with PBS/0.2% Triton and then incubated with 10 μL of TUNEL reaction mix at 37 °C for 60 min in a humid atmosphere. The slides were embedded in mounting medium prior to analysis. Fluorescence was examined under a Nikon Eclipse 80i fluorescence microscope at 488 nm. The percentage of TUNEL positive cells was then calculated using ImageJ [54].

Mitochondrial anion superoxide detection. Mitochondrial dysfunction was detected by fluorescence microscopy and quantified by flow cytometry using MitoSOX Red mitochondrial superoxide indicator (Invitrogen, ThermoFisher Scientific). For fluorescence microscopy, the cells were grown on 8-well chambered coverslips, while for flow cytometry analysis, the cells were seeded in 48-well plates. Cells were analyzed 1 h, 3 h and 6 h post-CAP treatments. Two µM antimycin A (Sigma-Aldrich) was used as positive control for mitochondrial superoxide anion production. Treated cells were immersed in 100 nM MitoTracker Green (Invitrogen, ThermoFisher Scientific) to label the mitochondrial network, and in 1 μM of MitoSOX red reagent solution, and incubated for 20 min at 37 °C, protected from light. Cells were observed in live microscopy using a confocal Leica SP8 microscope. ImageJ software (KW Eliceiri, US National Institutes of Health, Bethesda, MD, USA) was used to process the fluorescence images. For flow cytometry, the cells were washed twice with cold PBS and resuspended in FACS buffer (1 × PBS, 2% bovine serum albumin (BSA)). Fluorescence intensity was measured by flow cytometry with an excitation wavelength of 510 nm and an emission wavelength of 580 nm. Data were processed with FlowJO Software version v10.7.1 (BD Life Sciences, Franklin Lakes, NJ, USA).

Detection of mitochondrial membrane polarization. Changes in mitochondrial membrane potential were analyzed by fluorescence microscopy, and quantified by flow cytometry using tetramethylrhodamine, methyl ester (TMRM) dye (Invitrogen, ThermoFisher Scientific). Cells were analyzed as described above, except that they were incubated in 100 nM MitoTracker Green and 100 nM of TMRM reagent solution for 30 min at 37 °C. We used 2 µM carbonyl cyanide m-chlorophenyl hydrazone (CCCP, Sigma-Aldrich) for 30 min as a positive control for membrane depolarization.

Detection of intracellular ROS. The amount of intracellular ROS was quantified by flow cytometry using green CellROX^®^ Oxidative Stress Reagent (Invitrogen, ThermoFisher Scientific). Cells were analyzed as described above, except that they were incubated in 500 nM Green Cell ROX reagent solution for 1 h at 37 °C.

Immunofluorescence microscopy. Keratinocytes were grown on glass coverslips for 24 h, and plasma treated as described above. At appropriate time points, the cells were washed twice with PBS, fixed with 4% (*w*/*v*) paraformaldehyde for 15 min, permeabilized with PBS/0.2% Triton for 5 min, and treated with PBS/2% BSA/0.2% Triton for 1 h. The slides were incubated at RT for 2 h with anti-HSP27 (Cell Signaling Technology, Danvers, MA, USA; clone G31, 1:200) and anti-Ki67 (Abcam, Cambridge, UK; ab16667, 1:200) antibodies in PBS/2% BSA/0.2% Triton. Next, the cells were washed twice with PBS/2% BSA/0.2% Triton and incubated for 1 h with secondary antibodies, Alexa Fluor 533-conjugated goat anti-mouse IgG and Alexa Fluor 488-conjugated goat anti-rabbit IgG, respectively (Invitrogen, 1:1000). Coverslips were washed again with PBS and mounted using Mowiol^®^ (Sigma-Aldrich) with 4′,6-diamidino-2-phenylindol (DAPI). Samples were observed using a Nikon Eclipse 80i fluorescence microscope coupled to a Nikon DXM1200C camera.

Scratch assay. Keratinocytes were seeded (25,000 cells/well) into 96-well plates, cultured for 24 h until complete confluence, and 800 µm wide standardized scratches were made using the WoundMaker Tool (Sartorius, Gottingen, Germany). The wounded cells were washed twice with culture medium to remove any detached cells and then treated with CAP for the indicated periods of time, covered with 200 µL/well of PBS corresponding to a 6 mm thick PBS layer. Immediately after plasma the treatments, images of the plates were monitored by bright-field microscopy using the IncuCyte ZOOM System (Sartorius). Images of the wound were taken at hourly intervals for 24 h (at that time the wound area was completely closed in the control condition). Images were processed with the Incucyte^®^ Scratch Wound Analysis Software module. For the manual wound healing assay, scratches were made with a 200 µL tip on keratinocyte monolayers at complete confluence (scratches were 800 ± 125 µm wide). The wounded cells were washed twice and then treated with CAP for the indicated periods of time. Images of the scratches were taken by bright-field microscopy at 1 h, 24 h and 48 h post treatment. To inhibit cell proliferation, cells were pretreated with 5 µg/mL mitomycin C for 3 h.

Reconstructed Human epidermis. Reconstructed human epidermis (RHEs) were generated as previously described [36] using primary human keratinocytes. RHEs were harvested after 12 days of culture at the air-liquid interface [55].

Morphological and immunohistochemical analysis of RHEs. RHEs were fixed in 4% formalin solution at 4 °C overnight, and subsequently embedded in paraffin. Five μm thick sections were stained with hematoxylin and eosin (H&E, Sigma-Aldrich) or processed for immunohistochemical analysis. Sections were deparaffinized, unmasked in 50 mM glycine buffer for 30 min, blocked for 1 h with PBS/0.05% Tween/3% Fetal Bovine Serum (FBS), and incubated with rabbit anti-keratin 10 (Covance, Princeton, NJ, USA; 1:1000), mouse anti-filaggrin (homemade, clone AHF3, [56], 1:1000), mouse anti-involucrin (Labyrinth Bio-Pharma, Bloomington, IN, USA; 1:1000), and rabbit anti-corneodesmosin (Sigma-Aldrich, 1:500) antibodies diluted in PBS/0.05% Tween/3% FBS for 2 h at room temperature. After three washes, sections were incubated with secondary antibodies Alexa Fluor 488-conjugated goat anti-rabbit IgG, Alexa Fluor 533-conjugated donkey anti-mouse IgG or Alexa Fluor 488-conjugated goat anti-mouse IgG and observed as described for cells on coverslips.

Statistical analyses. At least three independent experiments were performed in all assays. Individual data are plotted and expressed as mean ± SD. Data analysis was performed using GraphPad Prism 9 program (GraphPad Software, Inc., San Diego, CA, USA). If not indicated otherwise in the figure legends, the data were statistically analyzed using two-way analysis of variance (ANOVA) followed by Dunnett’s multiple comparisons post-test to compare each condition with the control.

## Figures and Tables

**Figure 1 ijms-23-10650-f001:**
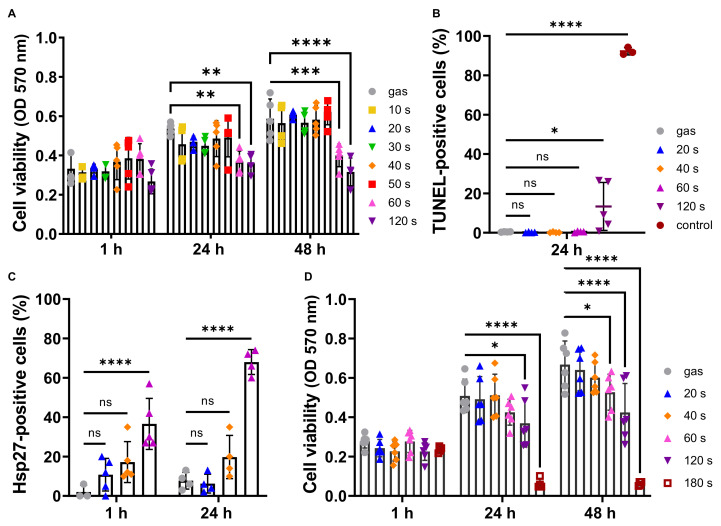
CAP treatments of less than 40 s did not affect keratinocyte proliferation and viability. N/TERT-1 keratinocyte cell line (**A**–**C**) and human primary keratinocytes (**D**) were exposed to CAP for the indicated periods of time. Cell viability was then determined using an MTT assay 1 h, 24 h and 48 h after CAP exposure (**A**,**D**), with cells treated with gas alone for 120 s as control; n ≥ 3. Induction of N/TERT-1 apoptosis was evaluated with a TUNEL assay 24 h after CAP treatments (**B**), 1 µM staurosporine being used as positive control; n = 3. Induction of heat stress was determined by immunocytochemistry on N/TERT-1 keratinocytes using an anti-HSP27 antibody, 1 h and 24 h after CAP exposure (**C**); n = 3. Error bars represent S.D.; *, *p* < 0.05; **, *p* < 0.005; ***, *p* < 0.001; ****, *p* < 0.0001; ns: not significant.

**Figure 2 ijms-23-10650-f002:**
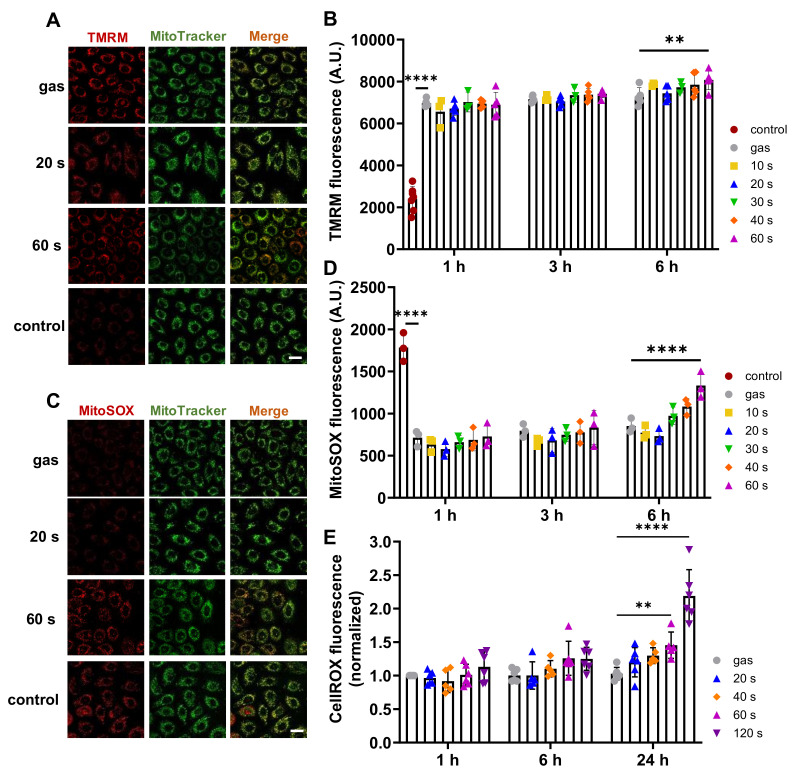
CAP treatment of less than 60 s did not affect N/TERT 1 mitochondria and intracellular ROS. N/TERT-1 cells were exposed to CAP for 10–60 s. Putative changes in mitochondrial membrane potential were determined by TMRM fluorescence assay 1 h, 3 h and 6 h post-treatment. Representative confocal microscopy images are shown (**A**) and fluorescence intensities were quantified by flow cytometry (**B**). We used 2 µM CCCP as positive control for membrane depolarization. n = 3. Mitochondrial dysfunctions were detected using the MitoSOX Red fluorescence assay. Representative confocal microscopy images are shown (**C**) and fluorescence intensities were quantified by flow cytometry (**D**). Two µM antimycin was used as positive control for mitochondrial anion superoxide production. n = 3. Changes in intracellular ROS levels were detected using green CellROX reagent by flow cytometry (**E**). Scale bars correspond to 20 µm. Error bars represent S.D.; **, *p* < 0.005; ****, *p* < 0.0001; ns: not significant.

**Figure 3 ijms-23-10650-f003:**
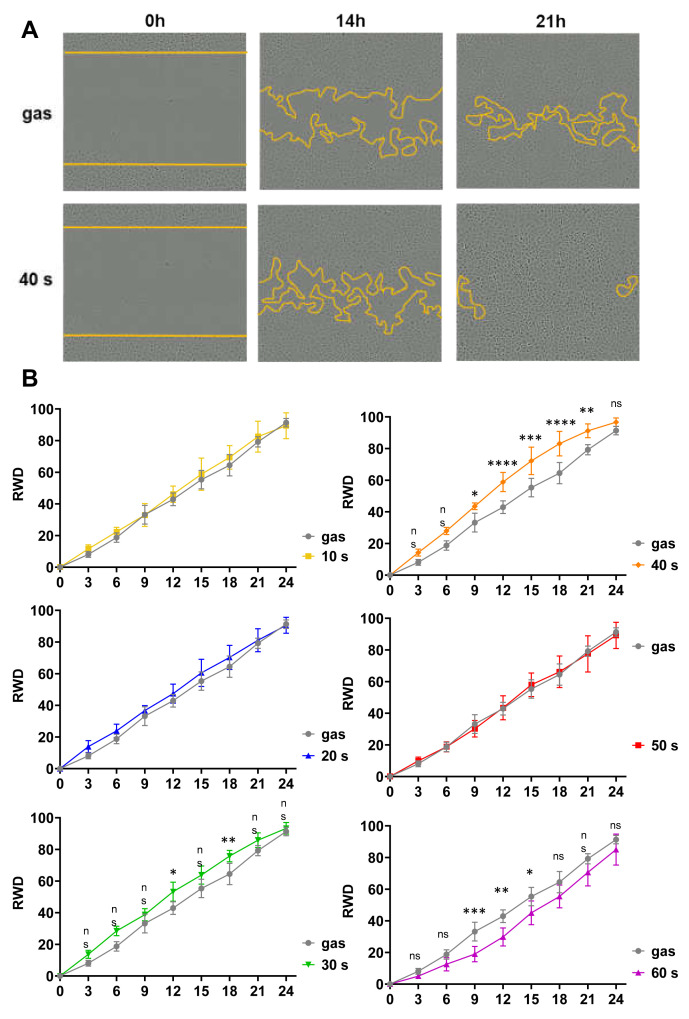
Short-time CAP treatments of N/TERT-1 increased cell migration. Standardized scratches were made on a completely confluent cell monolayer, cells were exposed to CAP for between 10 s and 60 s, and images of the wound area were taken at hourly intervals for 24 h to quantify cell migration. Treatments with gas alone for 60 s was used as control. (**A**). Representative images of those taken immediately after the scratches (0 h), and 14 h and 21 h later. Yellow lines indicate the edges of the wound. (**B**). Quantitative analyses of scratch fillings were performed, and the results are expressed as relative wound density (RWD). The results corresponded to the measurements in 6 separate wells; n = 6, error bars represent S.D.; *, *p* < 0.05; **, *p* < 0.005; ***, *p* < 0.001; ****, *p* < 0.0001; ns: not significant.

**Figure 4 ijms-23-10650-f004:**
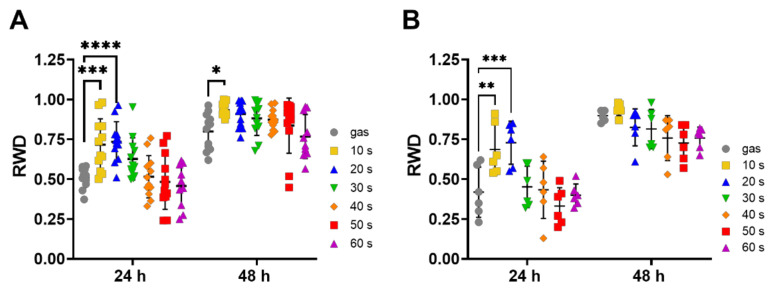
Short term CAP treatments of human primary keratinocytes stimulated their migration. Scratches were made manually on a completely confluent monolayer of primary human keratinocytes pre-treated for 3 h (**B**) or not (**A**) with 10 µg/mL mitomycin, and cells were exposed to CAP for 10 to 60 s, as indicated. The plate wells were monitored by bright-field microscopy 24 h and 48 h after plasma treatments, and quantitative analyses of the scratch closing, expressed as relative wound density (RWD), were performed. Treatments with gas alone for 60 s were used as control. The results corresponded to experiments performed with keratinocytes from 3 different donors; n = 6, error bars represent S.D.; *, *p* < 0.05; **, *p* < 0.005; ***, *p* < 0.001; ****, *p* < 0.0001; ns: not significant.

**Figure 5 ijms-23-10650-f005:**
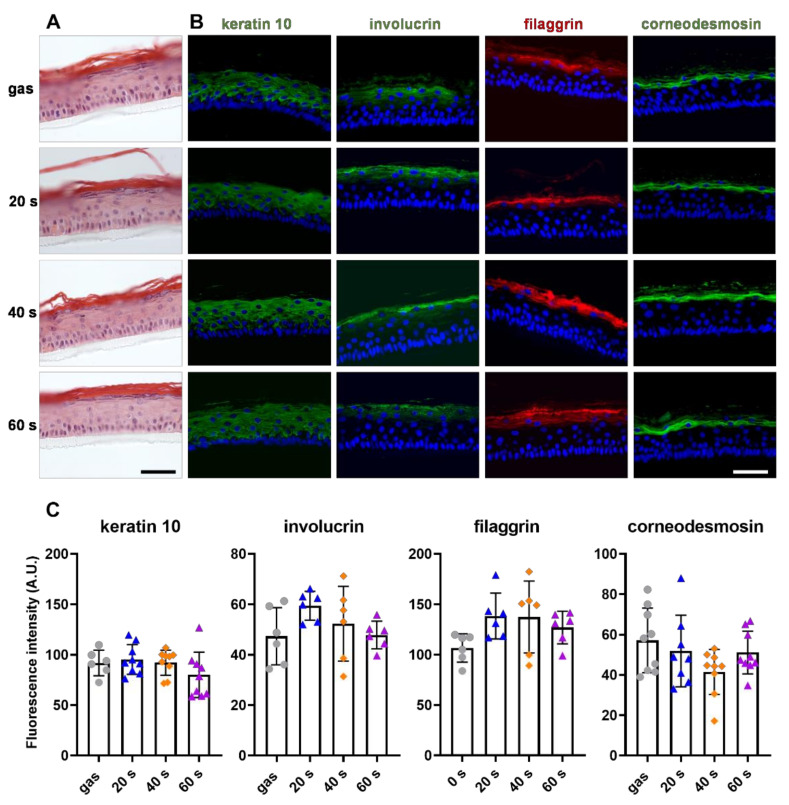
CAP treatments did not affect primary keratinocyte differentiation. Human primary keratinocytes were exposed to the direct plasma jet for 20 s, 40 s and 60 s, and used to produce reconstructed epidermis at the air-liquid interface; treatment for 60 s with gas alone was used as control. The resulting epidermis were formalin-fixed and paraffin-embedded. (**A**–**C**) Five µm sections were stained with H&E (**A**) and immunodetected with antibodies specific for keratin 10 (green), involucrin (green), filaggrin (red) and corneodesmosin (green) (**B**,**C**). Representative immunofluorescence images are shown in (**B**). Nucleus staining (DAPI) is in blue. Scale bars are 50 µm. Immunofluorescence signals were quantified using ImageJ (**C**). Error bars represent S.D.

## Data Availability

All data are contained in this article.

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
