# Peer review of "Cold Atmospheric Plasma Jet Treatment Improves Human Keratinocyte Migration and Wound Closure Capacity without Causing Cellular Oxidative Stress"

_ijms, 2022, doi:10.3390/ijms231810650_

Round 1

Reviewer 1 Report

See Attached review

Reviewer 2 Report

The authors investigate an important issue related the wound healing effect of plasma treatment. Couple of points may be still clarified in the presentation.

1. Related to the plasma source. Under the applied conditions what is the length of the active plasma? What species are expected to reach the surface of the PBS. This may be a good comparison with the other sources. It could be also commented on the existence/non-existence of electrical field and temperature effects.

2. The treatment conditions and protocols. What is the thickness of the PBS layer covering the cells during treatment. What is the expected diffusion time of the active species to the cells. This can be important for the very short treatments. The authors claim that the treated PBS is changed immediately after the treatment. In the case of the 5s treatment how much elapsed time this could mean? Do authors expect any active species interaction with the cells during the very short treatments? It would be important to give an estimated interaction time for every treatment condition.

If the treated PBS is immediately changed, how this model can be related to the real life scenario, when  the wound covered by body fluids/blood is continuously covered after the treatment.

3. Previous studies have also used human keratinocytes, namely immortalized human keratinocytes in the case of a He plasma needle. Authors may refer to this study in the introduction and the discussion. Korolov et al. J. Phys. D: Appl. Phys. 49 (2016) 035401 (12pp) . In that study a very exact method has been used to measure the cell proliferation, namely xCELLigence microelectronics biosensor, while the effect of the plasma needle on wound closer has been also evaluated.
